# The Predisposition of Men Who Have Sex with Men to Use Post-Exposure Prophylaxis for HIV in a Capital City in Northeast Brazil

**DOI:** 10.3390/ijerph22020210

**Published:** 2025-02-02

**Authors:** André Felipe de Castro Pereira Chaves, Yndiara Kássia da Cunha Soares, Eugênio Barbosa de Melo Júnior, Rosilane de Lima Brito Magalhães, Shirley Verônica Melo Almeida Lima, Paulo de Tarso Moura Borges, Telma Maria Evangelista de Araújo

**Affiliations:** 1Postgraduate Program in Nursing, Nursing School, University of Piauí, Recife 64049-550, Brazil; yndiarakassia@hotmail.com (Y.K.d.C.S.); eugeniobmj@gmail.com (E.B.d.M.J.); rosilane@ufpi.edu.br (R.d.L.B.M.); telmaevangelista@gmail.com (T.M.E.d.A.); 2Nursing Department, Federal University of Sergipe, Lagarto 49400-000, Brazil; shirleymelo.lima@gmail.com; 3Specialized Med Department, Universidade Federal do Piauí, Teresina 64049-300, Brazil; ptborges@gmail.com

**Keywords:** sexual and gender minorities, post-exposure prophylaxis, HIV

## Abstract

The aim of this study was to analyze the predisposition and factors associated with the use of Post-Exposure Prophylaxis (PEP) for HIV in men who have sex with men (MSM). This was a cross-sectional study conducted in the city of Teresina, Piauí, Brazil, between January and July 2024. The study sample consisted of 320 MSM. A questionnaire consisting of 37 previously validated questions and a risk perception scale for HIV with eight questions were used. To explain which factors would be associated with predisposition to the use of PEP, a logistic regression analysis was applied with an odds ratio. The criterion for including variables in the logistic model was an association at the 20% level (*p* ˂ 0.20) in the bivariate analysis. Statistical significance in the final model was set at 5%. Although the vast majority of MSM reported a willingness to use PEP (94.4%), their knowledge about prophylaxis and their HIV risk perceptions were largely unsatisfactory. It was found that living alone reduces the chances of predisposition to PEP use by 75% (AOR = 0.25; *p* = 0.01), and using a condom during oral sex reduces the chances of predisposition to PEP use by 91% (AOR = 0.09; *p* < 0.001). In light of this, the importance of greater investments in health education actions that reinforce the mechanisms of HIV transmission, as well as the use of methods for its prevention, is highlighted. In addition, targeted interventions are needed to improve knowledge about PEP and HIV risk perception.

## 1. Introduction

Human Immunodeficiency Virus (HIV) and its global spread are considered public health problems, requiring increasingly effective responses to prevent its transmission. In 2022, 39 million people were living with HIV worldwide, while in Brazil, during the same period, this number corresponded to just over 1 million cases. In the state of Piauí, in 2022, there were 516 new cases of the infection, representing a variation of a 1% increase when compared to the previous year [1].

The HIV epidemic affects populations disproportionately, with some groups being considered more vulnerable than others, such as men who have sex with men (MSM) [2]. In Brazil, while the prevalence of HIV was estimated to be 0.6% in the general population, the estimate for MSM was 18.4%; thus, it is considered an epidemic that has lasted for decades in this group [3].

The public considered to be a key population in HIV transmission includes gay individuals, bisexual individuals, sex workers, people who use alcohol and other drugs, transgender individuals, and people deprived of liberty. Despite their importance in the dynamics of infection by the immunodeficiency virus, research on this specific group is still scarce, and there is low government investment in this population group [4].

The new global HIV elimination targets demand more than the progress achieved so far, requiring 95% of individuals at risk of contracting HIV to adopt combined prevention strategies, along with the new testing and treatment goals aimed at achieving the “95-95-95” target set by the World Health Organization across all subpopulations, age groups, and geographic contexts [5]. This goal focuses on reducing the inequalities that drive the AIDS epidemic and prioritizing individuals who still lack access to HIV services.

A U.S. study revealed that the lack of knowledge about PEP among potential users contributes to its underutilization. Failure to adhere to this measure within the biomedical model ultimately increases the likelihood of individuals to be infected with HIV [6].

The Ministry of Health (MS) adopts Post-Exposure Prophylaxis (PEP) as one of the strategies of the prevention model. This measure consists of the uninterrupted use of combined antiretrovirals for 28 days, and its use should begin within 72 h after exposure to the virus. PEP contributed to preventing, on average, the transmission of HIV in 3138 people from 2009 to 2017, which corroborates its high standard of effectiveness [7]. In Piauí, between 2018 and 2024, a total of 8638 PEP dispensations were recorded. However, despite the increase in HIV incidence in the state, there was no significant upward trend in PEP dispensation over this period.

A study carried out in Rio Grande do Sul, Brazil, showed that gay men and MSM were prevalent in seeking PEP compared to individuals of other sexual orientations, indicating that the majority reported multiple sexual partners [8]. Furthermore, not using condoms during sexual relations can also trigger a perception of a risk of contracting HIV, which, to a certain extent, leads to an increase in the demand for PEP [9]. Given the sociocultural differences between regions, it is important to analyze the predisposition to PEP in other regional contexts.

Although the trend of PEP use is increasing among MSM, some individuals still lack the necessary knowledge about prophylaxis, which becomes a barrier and delays the search for healthcare services. Therefore, the dissemination of information on the internet was one of the main means to search for knowledge about PEP [10].

In addition to further exploring the sexual practices that may increase the vulnerability of MSM to HIV/AIDS, this study will contribute to identifying the predisposition of this key population to using PEP, with the potential to assist health managers in planning and implementing public policies that include more effective programs for MSM as a vulnerable population at risk of HIV and other STIs.

Given the above, this study will be pioneering in estimating the prevalence of the predisposition to PEP use among MSM in response to risk exposures, as well as in identifying associated factors, considering that adherence may vary across different regions of the country. Thus, the present study seeks to analyze one’s predisposition to the use of PEP for HIV and associated factors in men who have sex with men (MSM) in Teresina—PI.

This study’s hypothesis is that socioeconomic factors, sexual behaviors, lifestyle, and knowledge about PEP influence the predisposition to its use among MSM.

## 2. Materials and Methods

This work is a cross-sectional study. It was prepared according to the guidelines of the Strengthening the Reporting of Observational Studies in Epidemiology (STROBE) instrument.

This study was conducted at the Testing and Counseling Center (CTS) in the city of Teresina, capital of Piauí, Brazil. In addition to the CTS, the study also included socializing environments for the target audience, such as squares, a university, and a library, in an attempt of capture different profiles of the MSM population.

The source population of the study consisted of MSM residing in the city of Teresina, corresponding to 14,705 [11]. To calculate the standard initial sample size, an assumed prevalence of 50% regarding the predisposition to the use of PEP in MSM was adopted, considering that there are no data in the national literature regarding this event in the study population, and this value maximizes the sample. A 95% confidence level was used, along with a tolerable error of 5.5% [12], resulting in a minimum sample of 320 MSM.

MSM who had at least one sexual relationship with another man in the past twelve months and were 18 years of age or older were included. The exclusion criteria were being under the influence of alcohol and/or drugs at the time of data collection, having an HIV+ status, and already using PEP or PrEP.

The dependent variable was the predisposition to PEP use with the outcomes of *yes/no* in response to the following question: “After a potential HIV exposure, would you be willing to use PEP to prevent infection with the virus?” The independent variables were sociodemographic characteristics, health conditions/information, sexual practices, knowledge about PEP, and perception of HIV risk.

Data collection was carried out from January to July 2024. Recruitment was carried out using non-probability snowball sampling, as this is a population that is difficult to access and identify [13].

Data collection procedures were carried out in two consecutive stages. In the first, the Rapid Test (RT) for HIV was performed, and post-test counseling was provided to MSM who agreed to take it. The RT was necessary to exclude HIV+ individuals from the study. However, it is important to emphasize that these individuals received counseling and were referred to the reference service to begin treatment. The second stage was carried out through interviews conducted with a form and the HIV risk perception scale.

The research instrument was a questionnaire subdivided into three parts. The first part contained twenty-six questions regarding sociodemographic data, sexual health, and sexual practices, adapted from an instrument validated by Almeida et al. (2023) [14]. The second part consisted of nine questions related to knowledge about PEP, based on an instrument previously validated by Matos et al. (2021) [15]; the third part included two questions related to the predisposition to use PEP, validated by Gomes (2017) [16]. Each correctly answered question about knowledge was worth one point. The maximum score obtained was nine points, corresponding to 100%. At the end, knowledge was classified as follows: unsatisfactory—1 to 69%; satisfactory—70 to 100% [15].

In addition, an HIV risk perception scale was applied with eight questions, the scores of which can vary from 1 to 6. The maximum score obtained on the scale is 40 points, which corresponds to 100%. Thus, risk perception was classified as follows: 1 to 69%—unsatisfactory; 70 to 100%—satisfactory [17].

It is noteworthy that, despite the instruments mentioned having been previously validated in a different context, in order to assess their adequacy and comprehension, a pilot test was conducted with 30 MSM who did not participate in the study analyses.

The data were organized in a database in Microsoft Excel software and later exported to the Statistical Package for Social Science (SPSS), version 26.0. The Kolmogorov–Smirnov test was performed to verify the assumption of normality of the distribution of quantitative variables. In the analysis of the predisposition to PEP use as a *yes/no* response variable, a logistic regression analysis was applied with crude odds ratio (COR), adjusted odds ratio (AOR), and respective confidence intervals, seeking to explain which factors would be associated. The criterion for including variables in the logistic model was an association at the 20% level (*p* ˂ 0.20) in the bivariate analysis [18]. The criterion of using *p* ≤ 0.2 in the bivariate analysis was adopted to ensure that potential predictors were considered at the initial stage, even if they were not highly significant in the univariate analysis. This wider limit was applied with the aim of avoiding a premature exclusion of variables that may become significant in an adjusted model [19]. The criterion for significance or permanence of the variables in the final model, in turn, was an association at the 5% level (*p* ˂ 0.05). The project was assessed and approved by the Research Ethics Committee of the Federal University of Piauí, logged under no. 6,428,674.

It is important to emphasize that all participants who agreed to take part in the study signed the Informed Consent Form (ICF). During data collection, confidentiality, privacy, image protection, non-stigmatization, and the non-use of information to the detriment of the participants were ensured.

## 3. Results

A total of 320 MSM were approached, with 41 being excluded due to PrEP use, 7 who were reactive to the HIV test, and 20 who refused to undergo testing, resulting in a final sample of 320 MSM, as planned. Considering that the instrument’s questions were understandable, there were no incomplete answers.

The MSM sample was predominantly in the 20–39 age group (90.6%), with the vast majority declaring themselves to be male (99.4%), homosexual (73.8%), with a family income of BRL 3500.00 (50%), with 18 years of education, and living with their parents or family members (63.9%) (Table 1).

With regard to sexual practices, most reported having sex with a condom in casual partnerships (68.8%); however, the majority had a steady partner (37.8%). The most commonly used method of protection against STIs was condom use (88.4%), followed by knowing one’s own serological status and that of one’s partners (28.1%). However, 5.3% reported not using any preventive measures. The majority reported using condoms during anal intercourse (95.6%), while during oral and vaginal intercourse, the percentages dropped to 10.9% and 14.1%, respectively, and 2.5% still reported not using condoms during any sexual intercourse (Table 1).

Regarding sexual practices, the majority practiced oral sex with a person of the same sex (86.9%), followed by casual anal sex with a person of the same sex (74.4%). The use of alcohol during sex was reported by 58.7% of the sample; however, the use of illicit drugs was denied by 83.1% of them (Table 1).

Most of the participants had unsatisfactory knowledge about PEP (84.4%), as well as an unsatisfactory perception of the risk of HIV (84.7%) (Table 2).

The vast majority of MSM declared a predisposition to PEP use (94.4%). When asked about the reasons for their lack of predisposition, the majority reported a fear of side effects (52.9%) and a fear of people thinking that the person is HIV+ (23.5%) (Table 3).

It was observed that satisfactory knowledge of PEP among MSM was statistically associated with a bisexual orientation (*p* = 0.150), monthly household income (*p* = 0.018), and years of education (*p* < 0.001) (Table 4).

Satisfactory HIV risk perception among MSM remained statistically associated with homosexual (*p* = 0.048) and bisexual orientations (*p* = 0.164), living with parents/family members (*p* = 0.034), monthly household income (*p* = 0.127), and years of education (*p* = 0.101) (Table 5).

Among the eight variables statistically associated with a predisposition to PEP use, in the bivariate analysis, only two remained in the multiple model, namely living alone, which reduces the chances of predisposition to PEP use by 75% (AOR = 0.25; CI 95%: 0.09–0.72; *p* = 0.01), and using a condom during oral sex, which reduces the chances of predisposition to PEP use by 91% (AOR = 0.09; CI 95%: 0.03–0.27; *p* < 0.001) (Table 6).

## 4. Discussion

To the best of our knowledge, this is the first study conducted in Brazil to analyze the predisposition to PEP use for HIV and associated factors in MSM. Although the vast majority of MSM (94.4%) expressed a predisposition to PEP use, when faced with a high-risk situation of contracting HIV, it was observed that their knowledge about prophylaxis and their perception of risk to HIV were mostly unsatisfactory. Living alone proved to significantly reduce the chances of predisposition to PEP use. Data in the literature that explain this association are still scarce. However, our findings are consistent with a study conducted in the People’s Republic of China, which indicated that social and family support is considered a significant factor affecting the acceptability of prophylaxis [20]. Having a support network formed by available family and friends is of fundamental importance in helping to overcome problems that will inevitably arise throughout life and acts as a protective factor. Family ties promote the autonomy of families within the community, strengthening their access to the intersectoral network, the social assistance protection network and benefits, as well as to socio-assistance programs and services. Building new bonds and creating a social support network are strategies that increase the possibility of expanding self-care beyond family boundaries [21].

A study conducted in the state of São Paulo on the adherence rate to PEP showed that a total of 56.7% of users adhered to prophylaxis. However, few users attend follow-up consultations. Factors such as work, use of private services, forgetfulness, and considering follow-up unnecessary were the main reasons for non-attendance at follow-up consultations [22].

Unsatisfactory knowledge about PEP, which predominated in the study sample, may be reflected in the low perception of HIV risk. This is because when an individual has limited knowledge about HIV and its prevention methods, vulnerability to it tends to increase, causing the risk of transmission to be perceived as non-existent or minimally perceived [23]. It is worth noting that knowledge alone is not responsible for changing the perception of HIV risk, since sociocultural, political, and economic aspects also establish a relationship with the adoption of sexual practices and the risk perceptions of individuals [24].

Despite scientifically correct knowledge and an appropriate attitude being considered precursors of healthy practices, it is possible to observe that there is not always a linear relationship between them, that is, they are not always predictive of behavior change, making it essential to investigate whether other intervening factors exist [15]. In this regard, our study found no statistically significant association between knowledge about PEP, HIV risk perception, and predisposition to use, which does not imply that such an association does not exist. This supports the idea that, beyond knowledge, behavior may be influenced by the social or behavioral context in which it is embedded.

Furthermore, negative reactions and prejudiced beliefs related to homosexuality, referred to as homonegativity, are quite common in Latin American countries, particularly in Brazil. The association of MSM with sin, immorality, promiscuity, unhappiness, and loneliness may contribute to an unsatisfactory HIV risk perception, as well as to non-adherence to prophylactic measures, in this population [25,26].

A study conducted in the Federal District showed that one of the greatest challenges of PEP is the knowledge of its existence, making it a concern that almost two-thirds of all people are unaware of prophylaxis, and among those who claimed to be aware of it, it was found that their knowledge was still very incipient [27]. Another Brazilian study also found a slightly different result from ours, indicating that the lack of knowledge about PEP is a factor that hinders its use. Moreover, the lack of training and preparation of professionals in healthcare services, homonegativity, and the stigma related to HIV, both by healthcare professionals and service users, constitute barriers to accessing PEP [25].

Fear of possible side effects caused by PEP is the main reason for not using prophylaxis. Fear of adverse events caused by medications leads to low adherence to prophylaxis, as indicated by a review study. Fear is still present, although studies have demonstrated the safety of PEP, which has few adverse effects. In addition, the low perception of the risk of infection and poor knowledge about prophylaxis are factors that may influence the lack of predisposition to use it [28].

Using a condom during oral sex reduces the chances of predisposition to PEP use by 91%. This occurs because oral sex, for many people, is not associated with a risk practice for HIV infection [29]. Given the low perception of risk associated with unprotected oral sex, MSM may perceive PEP as unnecessary. The lack of condom use is a reflection of the vulnerability experienced by the MSM group, as well as discrimination due to sexual orientation, which add up and have repercussions on this population’s limited access to health services and HIV prevention strategies [30]. Given that there are multiple factors, the self-perceived risk of HIV infection did not always correspond to sexual behavior [31].

Furthermore, the discrepancy between perceived HIV risk and real risk, particularly based on engagement in risky sexual behaviors, reveals the need for interventions to improve awareness of risky sexual behaviors and, therefore, the accuracy of HIV risk perception [32]. Condom use should be combined with other preventive methods and is essential, but not the only alternative, to prevent HIV-negative individuals who are at constant risk of becoming infected, such as MSM and sex workers.

In Brazil, combined prevention is adopted, a strategy that combines different HIV prevention methods according to the individual characteristics and stage of life of each person. The basic premise established is that comprehensive prevention strategies must concomitantly observe these different focuses, considering the specificities of the subjects and their contexts. PEP constitutes another preventive form to be included among the preventive methods used in combined prevention [33].

The sociodemographic profile of the sample was similar to that found in other studies [34,35], in which there was a predominance of young, homosexual adults with a high level of education. A higher level of education may indicate greater access to information and, consequently, a greater connection with health services, contributing to an increased predisposition to prophylaxis [36]. However, this profile was not associated with the perception of HIV risk. It is important to understand that people with high levels of education may differ in unobservable aspects. From this perspective, while a satisfactory perception of risk and good health practices, for example, may be influenced by education, other unobservable factors may interfere, causing an opposite effect. Pereira et al. (2022) found opposite findings in their study when they observed that a reduced perception of HIV risk can be explained by a lower level of education [37].

Another interesting finding is that more than half of the participants use condoms with casual partners, and the majority have a steady partner. Studies have shown that among the factors for not using condoms is having a steady sexual partner. This points to an increase in HIV transmission rates [38,39].

One study conducted in nine Portuguese-speaking countries concluded that increased trust in a partner results in a higher rate of low-efficacy protective measures being used, in addition to the practice of using alcohol and other drugs more frequently and challenging sexual practices, which results in greater exposure to HIV [40]. This behavior is explained by insecurity at the beginning of the relationship, which makes the couple more likely to protect themselves. However, trust and familiarity with the partner increase and the perception of risk decreases as the relationship becomes more solid, causing those involved to reduce the frequency of use or stop using condoms altogether in sexual relations with a regular partner [41]. Condoms were the most widely used method of prevention against HIV in the study sample, followed by testing and knowledge of their partners’ serological status, which is consistent with the findings of another national study [10]. A study conducted in Northeastern Brazil indicates that testing normally only occurs after betrayal or the end of a relationship [42]. Given this, it should be noted that the ideal of trust established in sexual partnerships can contribute to an increase in HIV infection.

Consistency in condom use is also associated with economic class, level of education, and access to health services [43]. Since individuals with higher levels of education prevailed in this study, this finding explains the high adherence to condom use during sexual intercourse.

Although less frequent, some participants reported non-penetrative sex as a preventive method against STIs. The risk of HIV transmission in this practice is low; however, it still exists. This is because this practice does not avoid other activities, such as unprotected oral sex or ejaculation in the oral cavity, which are associated with transmission risks [44].

It was observed that the vast majority adopt condoms as a barrier method during anal sex; however, they do not use them during oral and vaginal sex. This allows us to infer biased knowledge about the forms of HIV transmission. Although penetrative sex is associated with a greater chance of HIV infection [45], oral sex is considered one of the factors for inconsistent condom use in the population of MSM, making this population more vulnerable and with a lower perception of risk to HIV infection [46]. Furthermore, another study carried out in Spain revealed that the level of self-esteem of MSM, together with the absence of risk perception and the search for sexual sensations, are associated with risky behaviors [47].

Most of the sample had an unsatisfactory perception of the risk of HIV infection. This result is consistent with a survey conducted in a Brazilian capital in which only 9% of MSM interviewed had a satisfactory perception of the risk of HIV infection [48]. Another study conducted with MSM in three Brazilian cities, located in the southern and midwestern regions of the country, revealed that less than 5% of the interviewees considered themselves at a high risk for HIV infection [38]. This shows that the perception of HIV risk is considered fragile in different regional contexts. In addition, a large portion of the MSM population has secret sexual interactions due to social pressures, which end up exacerbating the risk during intercourse [49].

The low perception of HIV risk among MSM is worrisome, since it can be considered a barrier to control and prevention efforts, such as PEP, PrEP, and periodic HIV testing. In light of this, it can be stated that there is an urgent need to identify the vulnerability of the sexually active Brazilian population to HIV infection, especially in the subgroups at a higher risk, such as MSM, in addition to adopting interventions that can reformulate notions and perceptions of risk. The absence of the perception of risk, a high level of self-esteem, and a great search for sexual sensations constitute risk factors for risky sexual behavior [50].

A comparative study of sexual practices among MSM between 2009 and 2016 showed an increase in the adoption of unsafe sexual practices, such as multiple sexual partners, bareback sex (with anal penetration), and an increased use of illicit drugs [51]. The widespread availability of technology and treatment, such as PEP and PrEP, reinforce the ideal of medicalized intervention, reducing the potential effectiveness of preventive measures, such as pre- and post-test counseling, in addition to condom use [52].

Although to a lesser extent, some individuals reported using illicit drugs during sexual intercourse, this practice, known as chemsex, is related to a decreased perception of risk against STIs, including HIV, in the MSM population. The justification for the use of psychoactive substances is the improvement in the quality of sex, leading to decreased inhibition and increased sexual arousal and pleasure. Furthermore, chemsex contributes to an increase in the duration of sexual relations, the adoption of multiple partners, the non-use of condoms, and more challenging sexual practices [53,54].

Studies carried out in sub-Saharan Africa and in Granado, Spain, indicated that increased impulsivity leads to a decrease in the perception of risk, and identified the use of alcohol as one of the main contributors to precipitating actions [31,55].

In the state of Piauí, a study conducted with adolescents indicated that risky sexual behaviors may be influenced by cultural, social, and familial factors, which shape norms and perceptions regarding sexuality. Therefore, it is necessary to develop strategies that consider not only individual factors but also social contexts, cultural norms, and access to quality healthcare services [56].

The main limitation of this study was the sample’s resistance to undergo HIV testing, which was an inclusion/exclusion criterion. However, this was resolved without prejudice to the study after better information was provided. Furthermore, the low number of individuals not predisposed to the use of PEP is also considered a limitation, which makes the statistical analyses weaker.

The fact that predisposition was assessed based on a single question is also considered a limitation of this study. This is because self-reported predisposition to PEP use may not accurately reflect the actual predisposition to its use. The snowball sampling methodology is also a limitation, as this study used a convenience sample collected from individuals in a single city, which resulted in findings that cannot be generalized to all MSM in Piauí or Brazil.

For future studies, we also consider it necessary to evaluate the availability of PEP in other regions of the country and among other population groups, especially those that comprise priority and vulnerable populations at risk for HIV infection. Understanding the different regional and social contexts is crucial for the adoption of health measures tailored to each specific group and region.

## 5. Conclusions

Although the majority of MSM reported a predisposition to use PEP, their knowledge about prophylaxis and their perception of HIV risk were mostly unsatisfactory. It is worth mentioning that a high predisposition does not necessarily translate into actual use, and future interventions may address both social support structures and misconceptions about PEP.

Among the factors investigated in this study, only living alone and using a condom during oral sex were associated with PEP use, reducing the chances of predisposition to its use by 75% and 91%, respectively.

In light of this, greater investment is needed in health education actions that reinforce the mechanisms of HIV transmission, as well as the use of methods for its prevention. It is also essential to develop and implement health programs specifically adapted to MSM, addressing not only HIV prevention, but also drug use and mental health, with interventions adapted to the specific needs of this group. Furthermore, there is a need for more targeted interventions to improve knowledge about PEP and the perception of HIV risk.

## Figures and Tables

**Table 1 ijerph-22-00210-t001:** The sociodemographic data and sexual practices of MSM in the study. Teresina, PI, 2024 (*n* = 320).

VARIABLE	*n*	%
IDENTIFICATION		
Age range		
<20 years	15	4.7
20–39 years	290	90.6
40–59 years	15	4.7
Sex		
Male	318	99.4
Intersexual	2	0.6
Sexual orientation		
Homosexual	239	74.7
Bisexual	73	22.8
Pansexual	8	2.5
Who you live with		
Alone	63	19.7
Parents and/or family members	204	63.9
Colleague(s)/friend(s)	27	8.5
Partner	22	6.9
Other	3	0.9
SEXUAL PRACTICES		
Sex with a condom with an occasional partner		
Yes	220	68.8
No	100	31.2
Most common type of sexual partner		
Occasional/casual	118	36.9
Constant	121	37.8
Constant and occasional/casual	81	25.3
STI preventive measures:		
Use of a condom		
Yes	283	88.4
No	37	11.6
Interrupted sex		
Yes	12	3.8
No	308	96.2
Sex without penetration		
Yes	57	17.8
No	263	82.2
PrEP		
Yes	7	2.2
No	313	97.8
Testing/knowing my HIV status and that of my partners		
Yes	90	28.1
No	230	71.9
No prevention measure		
Yes	17	5.3
No	303	94.7
Type of sexual relation using a condom		
Oral		
Yes	35	10.9
No	285	89.1
Anal		
Yes	306	95.6
No	14	4.4
Vaginal		
Yes	45	14.1
No	275	85.9
I never use a condom		
Yes	8	2.5
No	312	97.5
Sexual practices in the past 12 months:		
Oral sex with a person of the opposite sex		
Yes	46	14.4
No	274	85.6
Oral sex with a person of the same sex		
Yes	278	86.9
No	42	13.1
Casual anal sex with a person of the same sex		
Yes	223	69.7
No	97	30.3
Active anal sex with a person of the same sex		
Yes	238	74.4
No	82	25.6
Casual anal sex with a person of the opposite sex		
Yes	6	1.9
No	314	98.1
Active anal sex with a person of the opposite sex		
Yes	13	4.1
No	307	95.9
Vaginal sex with a person of the opposite sex		
Yes	27	8.4
No	293	91.6
Use of achohol before having sex		
Yes	90	28.1
No	132	41.3
Sometimes	98	30.6
Use of an illicit drug before having sex		
Yes	26	8.1
No	266	83.1
Sometimes	28	8.8
Age (median) = 25 years Monthly family income (median) = BRL 3.500,00 Current minimum salary = BRL 1412.00 Years of study (median) = 18		

Source: direct study.

**Table 2 ijerph-22-00210-t002:** Classification of knowledge about PEP and perception of risk of HIV from sample of MSM. Teresina, PI, 2024 (*n* = 320).

Variables	*n* (%)	95% CI
Classification of Knowledge		
Unsatisfactory	270 (84.4)	(15.4–60.4)
Satisfactory	50 (15.6)	(7.6–39.7)
Perception of Risk		
Unsatisfactory	271 (84.7)	(8.3–80.4)
Satisfactory	49 (15.3)	(1.6–11.7)

Source: direct study; 95% CI—95% confidence interval at a 5% significance level.

**Table 3 ijerph-22-00210-t003:** Predisposition to PEP use among MSM from study and reasons not to use it. Teresina, PI, 2024 (*n* = 320).

Predisposition to PEP	*n* (%)	CI-95%
Yes	302 (94.4)	(3.5–8.6)
No	18 (5.6)	(91.4–96.5)
**Reasons for not taking PEP**		
Fear of side effects	9 (52.9)	(30.3–74.6)
Fear of people thinking that I am HIV positive	4 (23.5)	(8.5–46.7)
Because I don’t like taking pills	3 (17.6)	(5.2–40.0)
Others	1 (5.9)	(0.6–24.4)

Source: direct study; 95% CI—95% confidence interval at a 5% significance level.

**Table 4 ijerph-22-00210-t004:** Association between classification of satisfactory knowledge about PEP and sociodemographic variables of MSM residing in Teresina, PI, 2024 (*n* = 320).

Variables	Bivariate Analysis			Multivariate Analysis		
	COR	CI-95%	*p*-Value	AOR	CI-95%	*p*-Value
Biological sex						
Male	-	-	0.999			
Intersexual	1					
Age group						
<20 years	-	-	0.999			
20–39 years	1.289	(0.282–5.898)	0.743			
40–59 years	1					
Sexual orientation						
Homosexual	0.528	(0.134–2.079)	0.361	0.484	(0.11–2.107)	0.334
Bisexual	0.328	(0.072–1.495)	0.150	0.364	(0.072–1.827)	0.219
Pansexual	1			1		
Who do you live with						
Parents and/or family	0.724	(0.328–1.597)	0.424			
Alone	0.792	(0.302–2.081)	0.637			
Other	1.000					
Monthly family income (mean ± SD) = 9234.32 ± 17,316.92; COR = 1.000; CI 95% = 1; *p* = 0.018; AOR = 1.00; CI 95% = 1; *p* = 0.196. Years of schooling (mean ± SD) = 21.12 ± 5.98; COR = 1.170; CI 95% = 1.095–1.250; *p* < 0.001; AOR = 1149.000; CI 95% = 1.074–1.230; *p* = 0.00.						

Source: direct research. COR = crude odds ratio. AOR = adjusted odds ratio. *p*-value: Wald test for logistic regression.

**Table 5 ijerph-22-00210-t005:** Association between satisfactory HIV risk perception and sociodemographic variables of MSM residing in Teresina, PI, 2024 (*n* = 320).

Variables	Bivariate Analysis			Multivariate Analysis		
	COR	CI-95%	*p*-Value	AOR	CI-95%	*p*-Value
Biological sex						
Male	5.625	(0.346–91.467)	0.225			
Female	1					
Age group						
<20 years	2.364	(0.361–15.455)	0.369			
20–39 years	1.132	(0.247–5.192)	0.874			
40–59 years	1					
Sexual orientation						
Homosexual	0.275	(0.076–0.991)	0.048	0.240	(0.062–0.923)	0.038
Bisexual	0.379	(0.097–1.488)	0.164	0.310	(0.074–1.293)	0.108
Pansexual	1			1		
Who do you live with						
Parents and/or family	3.739	(3.739–1.106)	0.034	4.024	(1.175–13.778)	0.027
Alone	2.376	(0.597–9.459)	0.220	2.130	(0.523–8.681)	0.292
Other	1			1		
Monthly family income (mean ± SD) = 4303.22 ± 3505.02; COR = 1.000; CI 95% = 1; *p* = 0.127; AOR = 1.00; CI 95% = 1; *p* = 0.209. Years of schooling (mean ± SD) = 16.84 ± 4.73; COR = 0.949; CI 95% = 0.890–1.010; *p* = 0.101; AOR = 0.953; CI 95% = 0.889–1.022; *p* = 0.176.						

Source: direct survey. COR = crude odds ratio. AOR = adjusted odds ratio. *p*-value: Wald test for logistic regression.

**Table 6 ijerph-22-00210-t006:** Logistic models for predisposition to PEP use by resident MSM in Teresina, PI, 2024 (*n* = 320).

Variables	Bivariate Analysis				Multivariate Analysis			
	COR	95% CI		*p*-Value	AOR	95% CI		*p*-Value
		LI	LS			LI	LS	
**IDENTIFICATION**								
Age group								
<20 years	1.00	0.00	-	1.00				
20–39 years	0.00	0.00	-	0.99				
40–59 years	1							
Average years of education	0.93	0.81	1.06	0.26	-	-	-	-
Biological sex								
Male	17.70	1.06	295.43	0.04	10.57	0.32	345.33	0.18
Intersexual	1				1			
Sexual orientation								
Homosexual	5.60	1.05	29.79	0.04	3.14	0.43	22.94	0.11
Bisexual	2.09	0.37	11.68	0.39	1.11	0.14	8.74	0.25
Pansexual	1				1			0.91
Who do you live with								
Parents and/or family	1				1			
Alone	0.24	0.09	0.66	0.006	0.25	0.09	0.72	0.01
Other	2.08	0.25	17.02	0.494	1.91	0.23	15.84	0.54
SEXUAL PRACTICES								
Sex with a condom								
with a casual partner	1							
Yes	0.03	0.09	0.99	0.04	4.44	0.81	24.28	0.08
No								
Type of most frequent sexual partnership	1			0.92				
Eventual/casual	1.20	0.39	3.70	0.74				
Steady	0.95	0.29	3.13	0.94				
Steady and casual/casual								
STI preventive measures	1			0.11	1			
None	3.77	0.94	15.12	0.06	2.34	0.37	14.76	2.34
Condom	5.03	1.01	24.91	0.04	6.00	1.03	34.91	6.00
Other								
Type of sexual intercourse in which a condom is used								
Oral	0.13	0.04	0.40	<0.001	0.09	0.03	0.27	<0.001
Yes	1				1			
No								
Anal	2.70	0.39	18.37	0.11	1.57	0.30	8.14	0.59
Yes	1				1			
No								
Vaginal	0.51	0.15	1.75	0.28				
Yes	1							
No								
I never use a condom	0.55	0.03	10.08	0.69				
Yes	1							
No								
Use of alcohol before	0.90	0.34	2.38	0.83				
having sex	1							
Yes								
No	0.50	0.17	1.47	0.21				
Classification of knowledge about PEP	1.08	0.30	3.89	0.90				
Satisfactory								
Unsatisfactory	1							
Risk perception	0.89	0.25	3.22	0.87				

Source: direct study. COR = crude odds ratio. AOR = adjusted odds ratio. LI = limit inferior. LS = limit superior. *p*-value: Wald test for logistic regression.

## Data Availability

The data can be consulted via the main author’s email.

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
