# Peer review of "The Predisposition of Men Who Have Sex with Men to Use Post-Exposure Prophylaxis for HIV in a Capital City in Northeast Brazil"

_ijerph, 2025, doi:10.3390/ijerph22020210_

Round 1
Reviewer 1 Report
Comments and Suggestions for Authors
I consider the article relevant and original. In this paper, the authors evaluated the factors associated with the use of PEP after exposure to the risk of HIV infection in a reasonable sample of Brazilian gay men. The study is methodologically adequate and well written. I suggest some points for reformulation, with the aim of improving it.
1. In the abstract, I suggest that the authors present the instruments used and the study design.
2. In the introduction, I consider it important to add the World Health Organization's goal of ending the HIV/AIDS epidemic (95%-95%-95%). I also suggest that the authors add a paragraph in the introduction, before presenting the study objectives, to clarify how the information from this study related to PEP may contribute to the production of knowledge that supports public health policies to reduce the incidence of HIV among gay men, a key population in confronting the HIV/AIDS epidemic.
3. The introduction fails to explain more clearly the effects of knowledge and adherence to PEP in reducing HIV incidence. Please insert a paragraph on that topic.
4. Please present a hypothesis for each objective in the introduction and return to each of them in the discussion of the results.
5. The method does not provide information on whether the sample is probabilistic. Please, add.
6. In the second paragraph of the method, there is a typo: “in attempt tp”.
7. I verified in the study results that it was an exclusion criterion (correctly) for participants who used PrEP, but this information is not included in the study method and needs to be added.
8. In the last sentence of the method, there is a translation error in the expression: “Opinion”. Please, fix.
9. In the results, I suggest that they present a correlation analysis between the variables knowledge about PEP, perception of risk regarding HIV infection and the most relevant sociodemographic variables, and that they present this result in a table, before the regression analysis table.
10. The lack of knowledge about PEP, found in more than 50% of the participants, is an alarming fact, and I suggest that the authors add this result to the study summary.
11. The discussion needs several reformulations to become more in-depth and adequately cover all the results found. At the beginning of the discussion, the authors mention that this is the first study on predisposition to PEP use among Brazilian gay men; however, they did not conduct a literature review that could support this statement; therefore, I suggest softening it, with an expression such as: “To the best of our knowledge, this is the first”
12. Still in the first paragraph of the discussion, there is a probable translation error, in the expression: People's Republic of China. Please review.
13. In the second paragraph of the discussion, I recommend that authors expand the discussion on the perception of HIV risk and knowledge about PEP, including the stigma related to HIV and the belief that is still very present in Brazilian culture, especially guided by religious beliefs, that understands HIV as punishment related to homosexuality, which reveals the double burden of stigma faced by gay men living with HIV, or who are at greater risk for infection (see more details at: https://doi.org/10.3390/ijerph21091167 and https://doi.org/10.3390/ejihpe14040070
14. I suggest that you add to the introduction and discussion a recent Brazilian study on conditions for adherence to PEP. In this study, the sample was not exclusively made up of gay men, but they were the majority: https://doi.org/10.3855/jidc.17515
15. In the discussion, the paragraph on fear of the adverse effects of PEP need to be further investigated. What does the review (article 7) reveal? Are there really reasons to avoid PEP even if you have had a high-risk HIV infection? The literature has consistently reported the safety of this therapy, which, if used in a timely manner, as should be done in the case of PEP, has very few adverse effects.
16. In the third paragraph of the discussion, I suggest that the authors add the characteristics of health services, homonegativity and HIV-related stigma among health professionals and internalized by the gay individual as barriers to access to PEP.
17. The authors mention in the discussion that there is a constant risk of HIV infection among serodiscordant couples. This information is incorrect. People living with HIV and who regularly using antiretroviral medication achieve an undetectable viral load, and undetectable = untransmittable; therefore, there is no risk of transmission (for more information, read PARTNER I and II studies).
18. An important limitation of this study is the low number of individuals unwilling to use PEP, which makes the statistical analyses more fragile – I suggest that this point be included in the study limitations (please insert the implications of it as well)
19. I suggest that the low knowledge about PEP and low perception of HIV risk be added to both the abstract and the conclusion.
20. I suggest that you also add directions for future studies in the area.
Best regards to the authors.
Author Response
LETTER TO THE EDITORIAL BOARD
Teresina, January 21, 2025.
Dear Editors and Reviewers,
We are grateful for the opportunity to revise our manuscript for submission to this prestigious journal. We have carefully reviewed all the comments and suggestions provided and have incorporated the majority of them.
For the few suggestions that were not implemented, we have provided justifications. All corrections have been highlighted in blue in the manuscript to facilitate review.
Reviewer 1
ABSTRACT
- The suggestion to include the study design and instruments used was implemented.
INTRODUCTION
- The World Health Organization's goal of ending the HIV/AIDS epidemic (95%-95%-95%) was added.
- A paragraph was added to clarify how this study's findings related to PEP contribute to public health policies aimed at reducing HIV incidence among gay men, a key population in the fight against the HIV/AIDS epidemic.
- A paragraph was added citing a study that links knowledge and adherence to PEP with reduced HIV incidence.
- The study hypothesis, aligned with the manuscript’s general and sole objective, was added and referenced in the discussion, as suggested.
METHOD
- The sample was explicitly described as non-probabilistic. Recruitment via the “snowball” technique was already mentioned, considering the difficulty of accessing this population.
- The typographical error (“tp”) was corrected.
- The use of PrEP as an exclusion criterion was added. This precaution had already been taken during the study, but the clarification is appreciated.
- A translation error in the term “Opinion” was corrected.
RESULTS
- Tables 4 and 5 were added to present associations between PEP knowledge, HIV risk perception, and sociodemographic variables.
- The manuscript summary now states that both knowledge about PEP and HIV risk perception were unsatisfactory.
DISCUSSION
- The phrase “As far as we know” was added at the start of the discussion.
- The translation of “People’s Republic of China” was reviewed.
- The discussion was expanded to address the influence of HIV stigma and religious beliefs in Brazilian culture on HIV risk perception and PEP knowledge.
- A paragraph was added discussing current PEP adherence trends, as mentioned in the introduction.
- Factors contributing to the non-use of PEP—such as low HIV risk perception and inadequate knowledge—were added. The safety of PEP therapy was also emphasized.
- The discussion now includes the lack of adequate training for health professionals and barriers created by homonegativity and HIV stigma.
- An incorrect argument regarding constant HIV infection risk among serodiscordant couples was removed.
- The small number of individuals unwilling to use PEP was discussed, along with its implications for the study’s limitations
- Unsatisfactory knowledge about PEP and HIV risk perception was emphasized in the abstract and conclusion.
- The suggestion to propose future research in this area was implemented.
Sincerely,
The Authors

Reviewer 2 Report
Comments and Suggestions for Authors
see attached

The English could be improved to more clearly express the research.
- Reason: While the language is generally understandable, certain sections (e.g., the discussion) use repetitive phrases and vague terminology.
- Actionable Feedback for Authors: Improve sentence structure and clarity, particularly in the results and discussion sections, to avoid redundancy and enhance readability
Author Response
LETTER TO THE EDITORIAL BOARD
Teresina, January 21, 2025.
Dear Editors and Reviewers,
We are grateful for the opportunity to revise our manuscript for submission to this prestigious journal. We have carefully reviewed all the comments and suggestions provided and have incorporated the majority of them.
For the few suggestions that were not implemented, we have provided justifications. All corrections have been highlighted in blue in the manuscript to facilitate review.
Reviewer 2
TITLE
- To avoid an overly long title, the study design was included in the abstract and methods section, adhering to the STROBE guidelines.
ABSTRACT
- The suggestion was incorporated.
- The suggestion to highlight findings in the abstract and conclusion was accepted.
INTRODUCTION
Background and Objectives
- A reference was added regarding the relationship between knowledge and PEP adherence, emphasizing that non-adherence increases susceptibility to infection.
- Updated data on PEP dispensation in Piauí (2018–2024) were included.
Literature Context
- The need to investigate PEP predisposition in other regional contexts was addressed.
- Discussion on regional disparities in PEP adherence, as noted in studies such as one from Rio Grande do Sul, was added.
METHOD
Study Design and Population
- A justification for the chosen locations was provided, as they represent known meeting points for MSM in the study’s target area.
- The concept of MSM was included to clarify the inclusion criterion.
Sampling and Recruitment
- The sample size paragraph was adjusted to clarify that it reflects a standard calculation for a non-probabilistic sample.
- Recruitment via the snowball method was reiterated.
Data Collection Instruments
- Sections and questions from validated instruments were clearly identified.
- Details were added regarding the scoring system (9 questions, each worth 1 point) and classification as satisfactory or unsatisfactory.
- A pilot test with 30 MSM (not included in the final analysis) was noted to verify instrument adequacy.
Analysis
- The use of p = 0.20 for confounding variables was justified, citing a reputable source.
- Logistic regression analyses, including crude (COR) and adjusted (AOR) odds ratios, were detailed.
ETHICS
- Ethical considerations, including informed consent and participant confidentiality, were described in the final paragraph of the methods section.
RESULTS
Description of the Population
- Variables were categorized dichotomously to streamline analysis.
Inferential Analysis
- Less relevant variables were removed as suggested.
- Confidence intervals were emphasized, and relevant discussions were incorporated into the discussion section.
DISCUSSION
Interpretation of Main Findings
- The role of family ties in accessing health services was reinforced.
- Discussion on low-risk perception associated with oral sex without a condom and its impact on PEP predisposition was included.
Comparisons with Other Regions
- A recent study on adolescents in Piauí, highlighting cultural and familial influences on risky behaviors, was added.
Knowledge and Risk Perception
- A paragraph was added to discuss the non-linear relationship between knowledge and behavioral change, as observed in this study.
- Although no significant association between PEP knowledge, risk perception, and predisposition was found, this finding was contextualized within broader behavioral factors.
LIMITATIONS
- The use of a single question to assess PEP predisposition was noted as a limitation.
- Snowball sampling methodology was highlighted as another limitation.
CONCLUSION
- The conclusion reaffirms factors associated with PEP predisposition. A statement emphasizing the need for targeted interventions to improve PEP knowledge and address misconceptions was added.
- We remain available to address any further questions or clarifications.
Sincerely,
The Authors

Round 2
Reviewer 1 Report
Comments and Suggestions for Authors
The authors have done a good job of revising the article. All the points mentioned in the first opinion have been met, so I suggest that the manuscript should be accepted for publication.
Author Response
Thank you for all the suggestions and improvements to the manuscript.
Reviewer 2 Report
Comments and Suggestions for Authors
-
Further Methodological Clarity
- If data on incomplete responses exist (e.g., participants who refused to answer specific questions but completed the rest), clarifying how those missing data were managed would help readers assess the dataset’s robustness.
-
Discussion of Confounders
- While the use of p < 0.20 as an inclusion criterion in the final model is standard, you might add one or two sentences explaining your rationale or referencing a key methodological paper to solidify the approach.
-
Broader Context on Oral Sex
- The finding that not using condoms during oral sex correlates with greater predisposition to PEP is intriguing. Providing a bit more detail on how risk perception might differ for oral vs. anal sex could strengthen the discussion and highlight potential public health messaging.
-
Minor Editorial Refinements
- A quick read-through by a native or near-native English speaker could address sporadic grammar or word choice issues in the introduction and discussion (e.g., “the fear of side effects was present” vs. “fear of side effects persists among participants”).
While the English is understandable, some sentences (particularly in the discussion) could be more concise. Minor editorial polishing and occasional rephrasing would enhance clarity
Author Response
LETTER TO THE EDITORIAL BOARD
Teresina, January 24, 2025.
Dear Editors and Reviewer (2),
We are grateful for the new opportunity to revise our manuscript.
FURTHER METHODOLOGICAL CLARITY
- If data on incomplete responses exist (e.g., participants who refused to answer specific questions but completed the rest), clarifying how those missing data were managed would help readers assess the dataset’s robustness.
Answer: Considering that the instrument's questions were understandable, there were no incomplete answers. This information was added to the first paragraph of the results.
DISCUSSION OF CONFOUNDERS
- While the use of p < 0.20 as an inclusion criterion in the final model is standard, you might add one or two sentences explaining your rationale or referencing a key methodological paper to solidify the approach.
Answer:
The rationale for including variables with p-value ≤ 0.2 in multivariate analysis is given to ensure that potential predictors are considered at the initial stage, even if they are not highly significant in univariate analysis. This wider limit aims to avoid premature exclusion of variables that may become significant in an adjusted model.
BROADER CONTEXT ON ORAL SEX
- The finding that not using condoms during oral sex correlates with greater predisposition to PEP is intriguing. Providing a bit more detail on how risk perception might differ for oral vs. anal sex could strengthen the discussion and highlight potential public health messaging.
Answer: Thanks for the comment. We added new explanation, in paragraph 8: “Given the multiple factors, the self-perceived risk of HIV infection did not always correspond to sexual behavior (Mbilizi et al., 2022). Furthermore, the discrepancy between perceived HIV risk and real risk, particularly based on engagement in risky sexual behaviors, reveals the need for interventions to improve awareness of risky sexual behaviors and, therefore, the accuracy of HIV perceptions. risk (Krajewski et al., 2024).”
MINOR EDITORIAL REFINEMENTS
- A quick read-through by a native or near-native English speaker could address sporadic grammar or word choice issues in the introduction and discussion (e.g., “the fear of side effects was present” vs. “fear of side effects persists among participants”).
Answer: The manuscript was translated from Brazilian Portuguese and revised into English by Todd Irwin Marshall, a native English speaker who has lived in Brazil for 25 years. The translation and revision certificate is attached.
Sincerely,
The Authors
